

# Universality and the thermoelectric transport properties of a double quantum dot system: Seeking for conditions that improve the thermoelectric efficiency

Ronald Santiago Cortes-Santamaria[1], J. A. Landazabal-Rodríguez[2],
Jereson Silva-Valencia[1], Edwin Ramos[3], Marcos Sergio Figueira[4]
and Roberto Franco Peñaloza[1]⋆

**1** Departamento de Física, Universidad Nacional de Colombia (UNAL),
A. A. 5997, Bogotá, Colombia
**2** Departamento de Ciencias Naturales,
Escuela Tecnológica Instituto Técnico Central (ETITC), Bogotá, Colombia
**3** Vicerrectoría de Investigaciones, Universidad Manuela Beltrán, Bogotá, Colombia
**4** Instituto de Física-Universidade Federal Fluminense (IF-UFF),
Av. Litorânea s/n, CEP:24210-346, Niterói, Rio de Janeiro, Brazil

⋆ rfrancop@unal.edu.co

## Abstract

Employing universal relations for the Onsager coefficients in the linear regime at the symmetric point of the single impurity Anderson model, we calculate the conditions under which the quantum scattering phase shift should satisfy to produce the asymptotic Carnot's limit for the thermoelectric efficiency. We show that a single quantum dot connected by metallic leads at the Kondo regime cannot achieve the best thermoelectric efficiency. We study a serial double quantum dot system, with the quantum dots immersed in ballistic conduction channels. Each QD exhibits a strong but finite local electronic correlation $U$. We show that maintaining one dot in the electron-hole symmetric point and allowing charge fluctuations in the other QD makes it possible to drive the system to the maximum thermoelectric efficiency. We also show that the bound states in the continuum (BICs) and quasi-BICs associated with the quantum scattering interference process drive the DQD to the maximum thermoelectric efficiency. We identify two types of quasi-BICs that occur at low and high temperatures: The first is associated with single Fano resonances, and the last is with several Fano processes. We also discussed possible temperature values and conditions that could be linked with the experimental realization of the results.

# 1  Introduction

The thermoelectric effects in conventional metals have been well-known since the beginning of the 19th century. These effects permit obtaining electricity by employing a temperature gradient (Seebeck effect - thermoelectric generators) or causing a temperature gradient using an electrical potential difference (Peltier effect - thermoelectric refrigerators). Unfortunately, in conventional metals, the thermal efficiency associated with these effects is very low due to the interdependent character of the electrical and thermal conductances. These effects allow the development of thermoelectric generators (TEGs) that only acquire some practical applications after the discovery that the doped semiconductor $Bi_2Te_3$ and its alloys $Sb_2Te_3$, and $Bi_2Se_3$ [1–3], present at room temperatures a higher dimensionless thermoelectric figure of merit ($ZT$) [1, 4] and a high power factor (PF) and, until now, dominate the commercial industry of TEGs [5].

Due to environmental pollution problems, TEGs have recently started to be used as a thermoelectric energy recovery to convert wasted heat into electric power. Two promising applications where the TEGs have been used are automotive energy recovery and hybrid solar energy converter systems. In the first case, the TEG is generally coupled to the exhaust gas system once the primary waste energy flows from this vehicle device. Gasoline vehicles present a better power output than diesel vehicles, and this energy could be used to power the car's electrical devices and improve engine performance [6]. In the second application, photovoltaic cells, only the photons with energies close to the cell band gap contribute to an effective electric conversion. The energy of the photons greater than the cell band gap is dissipated as heat. A hybrid solution involves attaching a TEG to the back of the cell to convert the wasted heat into electric power. Some experimental results show that the electric efficiency of the PV-TE system should be enhanced at around 8% when compared with the PV solar system alone [6].

Since the experimental realization of the single impurity Anderson model (SIAM) employing a quantum dot immersed into a two-dimensional electron gas (single electron transistor - SET) [7], the interest in the research on nanostructured devices has been growing continuously. The thermoelectric properties of a quantum dot (QD) in the presence of Kondo correlations were addressed in an experimental device in references [8,9]. The employment of nanoscopic systems as thermal rectifiers shows that it is possible to enhance the efficiency of macroscopic devices by controlling energy transport on a microscopic scale [10]. A recent

experiment measured the thermoelectric properties in the Kondo limit of a correlated QDs device [11], below and above the Kondo temperature, producing high-quality data that allows a quantitative comparison with numerical renormalization group (NRG) results [12]. Another recent experiment [13] realizes a thermoelectric conversion at temperatures around $T = 30K$ in $InAs/InP$ nanowire QDs by taking advantage of their strong electronic confinement. We can cite the following papers from the theoretical perspective [14–26] and others from the experimental point of view [11, 13, 27–31]. Furthermore, recent reviews can provide enriching insights into the topic [32–34].

In the theoretical calculations of this work, we neglect the phonon lattice contribution to thermal conductance. However, this approximation is perfectly justifiable due to the high degree of phonon emission process control attained recently by experimental realizations of the DQD geometry [35, 36]. In reference, [35], the authors have measured phonon emission rates in a GaAs/AlGaAs DQD; the isolation of the system from electronic reservoirs, the low temperatures, and the weak tunnel coupling compared to values of detuning allowed for a direct readout of the interaction spectral density of electron-phonon coupling. On the other hand, in reference, [36], the authors studied a serial DQD formed in an $InAs/InP$ nanowire coupled to two electron reservoirs. They obtained phonon-assisted transport, allowing the conversion of local heat into electrical power in a nanosized heat engine.

Bound states in the continuum (BICs) are generally valid for waves, including the wave functions of quantum mechanics. BICs remain localized even though they coexist with a continuum spectrum of waves that could dissipate energy [37]. Although this phenomenon was proposed in the context of quantum mechanics [38], it appears in many different classical and quantum systems [37, 39, 40]. BICs are localized states in the continuum spectrum with discrete energies or frequencies, invisible to manipulation once the transmittance does not exhibit its presence [41, 42].

On the other hand, Fano resonances appear when there is a quantum interference process in a system consisting of a continuous degenerated spectrum connected to a system that exhibits a discrete level spectrum [43]. The interference is produced among the electrons circulating along the system's two channels: the discrete levels and the continuous conduction bands. Generally, by detuning the system from the BIC conditions, for example, changing the hybridization between the leads and the dots, the BICs transform themselves to Fano resonances (quasi-BICs) [44, 45] exhibiting dips in the transmission spectra and sharp peaks in the density of states (DOS).

Theoretical calculations and DFT simulations [46–49] predict that Fano resonances formed crossing or near the chemical potential contribute to the increase of the $ZT$ parameter. This condition is only a realization of the Mahan-Sofo criteria [50] that predicts the best $ZT$ for the presence of a delta function in the density of states near the chemical potential. The reference [51] presents a critical discussion about this point. One example of this kind of system is the use of porphyrin-based molecules with different metal centers, like the family of metals Mn, Co, Ni, Cu, Fe, and Zn, connected to gold electrodes to form a single electron transistor (SET) [52–55].

The main result of this paper is to show that the optimal values of the phase shifts generated by the quantum scattering interference process in a double quantum dot system (DQD) produce maximum thermoelectric efficiency.

All calculations are based on the universality associated with the Onsager coefficients described in reference [56]. This universality is linked to a quantum scattering center modeled by the SIAM in the electron-hole symmetric condition, valid for the limit of low energy excitations associated with the Kondo effect. This universality permits us to obtain the effective quantum phase shifts that improve the thermoelectric efficiency. The other condition is a quantum interference process that permits achieving the effective quantum phase shifts required

and described in Sec. 3. When tuned by an external parameter like the hybridization or temperature, these phase shifts generate quasi-BICs in the density of states, which are responsible for the considerable increase of the dimensionless thermoelectric figure of merit $ZT$.

In Sec. 2, we define the model and the computation of the thermoelectric transport coefficients. In Sec. 3, we express the $ZT$ product in terms of the Mahan-Sofo parameter $\varepsilon$, employing the universal relations obtained in a previous paper [56]. Furthermore, we explore the quantum scattering process of getting the Carnot's machine limit for a single SET ($ZT \to \infty$, $\varepsilon \to 1$). In Sec. 4, we describe a system of DQDs, showing two conditions that create an effective quantum scattering phase shift that significantly improves the thermoelectric efficiency. In section 5, we present the discussion of the role of quasi-BICs linked to the enhancing process of $ZT$ at low temperatures. In section 6, we discuss the best conditions to obtain the enhancement of $ZT$ at high temperatures and the quasi-BICs generated by thermal excitations associated with rising multiple Fano resonances. Section 7 presents the conclusions and perspectives of this paper. Appendix A summarizes the cumulant Green's functions method employed in all the thermoelectric calculation properties. Appendix B contains a derivation of the thermoelectric transport properties and a short discussion of the efficiency power.

## 2 Model and theory

Fig. 1 presents a pictorial view of the single electron transistor. The Hamiltonian of the system can be written as

$$H = \sum_{\mathbf{k},\sigma} \sum_{\alpha=L,R} E^\alpha_{\mathbf{k},\sigma} c^{\alpha\dagger}_{\mathbf{k},\sigma} c^\alpha_{\mathbf{k},\sigma} + \left( E_d n_d + U n_{d\uparrow} n_{d\downarrow} \right) + \sum_{\alpha=L,R} \sum_{\mathbf{k},\sigma} \frac{V_\alpha}{\sqrt{2N}} \left( c^\dagger_{d\sigma} c^\alpha_{\mathbf{k},\sigma} + c^{\alpha\dagger}_{\mathbf{k},\sigma} c_{d\sigma} \right), \quad (1)$$

where the first term represents the conduction leads, characterized by free conduction electrons ($c$-electrons) to the right ($R$) and the left ($L$) of the QD (Fig. 1), the second term describes the QD, where $U$ represents the Coulomb repulsion between the electrons at the QD site and the dot energy $E_d$, controlled by the gate voltage [57,58], and the third term corresponds to the tunneling between the embedded dot and the left (L) and right (R) semi-infinite leads. The amplitude $V_\alpha$ is responsible for the tunneling between the QD and the lead $\alpha$. For simplicity, we assume symmetric junctions (i.e. $V_\alpha = V_L = V_R = V$) and identical leads (i.e., $E^L_{\mathbf{k},\sigma} = E^R_{\mathbf{k},\sigma} = E_{\mathbf{k},\sigma}$) connecting the QD to the quantum wire, which will be described by a structureless rectangular conduction band of width, $W = 2D$.

In this work, we employed the exact results obtained in an earlier paper of our group, using the NRG technique based on the universality associated with the Onsager coefficients (cf. Ref. [56]). In Sec. 3, we derive the dimensionless thermoelectric figure of merit $ZT$ using NRG. However, to obtain the results presented in the paper, we employed the cumulant Green's function method (CGFM) for the SIAM summarized in Appendix A [59]. The CGFM describes the different regimes of SIAM well: Empty orbital, mixed-valence, and Kondo, and it led to results that agreed with the NRG predictions. The Friedel's sum rule is satisfied numerically with an error percentage under the 1%. The method can be summarized as follows: 1. Diagonalize a Wilson chain composed of $N$ sites, here $N = 2$, with one impurity site and one

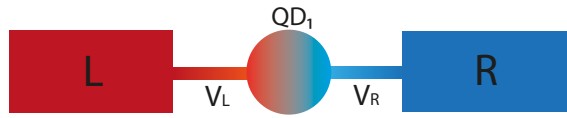

Figure 1: (Color online) Schematic picture of a quantum dot embedded into conduction leads (single electron transistor - SET).

conduction site. 2. Employing the Lehmann representation to calculate all the allowed atomic cumulant Green's functions (GFs). 3. Collect all the atomic GFs with non-zero residues and use them to calculate the atomic cumulants. 4. These atomic cumulants are approximations to calculate the full SIAM GFs.

# 3 Thermoelectric figure of merit and universality

In Appendix B, we derive the transport coefficients $L_0(T)$, $L_1(T)$, and $L_2(T)$, and the thermoelectric properties. In this section, we study the enhancement of the thermal efficiency of the SET, depicted in Fig. 1 and described by the Hamiltonian of Eq. 1, to produce the asymptotic Carnot's limit for the thermoelectric efficiency. We employed universal relations obtained as a function of the temperature normalized by the Kondo temperature $T^* = \frac{T}{T_K}$, and valid for the SIAM in the Kondo regime [56]. We investigated the dimensionless thermoelectric figure of merit $ZT$ as a function of $T^*$, associated with the thermoelectric efficiency as a function of the Mahan-Sofo parameter $\varepsilon$ [50]. The dimensionless thermoelectric figure of merit $ZT$ indicates the system's performance, is an indicator of the usefulness of materials or devices to be employed for thermopower generators or cooling systems. A summary of the evolution of $ZT$ values can be found in figure 1 of Vineis *et al.* review [60]. Nanoscopic systems open up new possibilities for increasing $ZT$, Coulomb interaction and level of quantization are the main "new elements" that could origin important changes in the thermoelectric properties. The $ZT$ is defined by

$$ZT = \frac{S^2 T G}{K + K_{ph}}, \tag{2}$$

where $G$ is the electrical conductance, $S$ is the thermoelectric power, $T$ is the absolute temperature [Eqs. B.4, B.7 and B.8]. If $ZT > 1$, the system could be a possible candidate for thermoelectric applications. $K_{ph}$ is the contribution of the phonons to thermal conductance that tends to decrease $ZT$; however, for simplicity, we do not consider the $K_{ph}$ contribution in this paper.

G.D. Mahan and J. O. Sofo [50] discussed the conditions a device must fulfill to produce the best thermoelectric efficiency, describing the thermoelectric transport properties in the linear regime. They investigated what type of electronic structure provides the most significant dimensionless thermoelectric figure of merit for thermoelectric materials. They determined that a narrow energy distribution of the carriers was needed to produce a large value of $ZT$.

Following the work of G.D. Mahan and J. O. Sofo [50], we define the Mahan-Sofo parameter $\varepsilon$ as a function of the thermoelectric coefficients, derived in Appendix B,

$$\varepsilon = \frac{L_1^2(T)}{L_o(T) L_2(T)}. \tag{3}$$

The Mahan-Sofo parameter $\varepsilon$ depends only on the thermoelectric coefficients, and it can be calculated employing the Boltzmann equation, the Landauer formalism, or any other pertinent method. In the original paper of Mahan and Sofo, they employed the Boltzmann formalism and used the transport distribution $\Sigma(\epsilon)$. In the present paper, we are interested in studying BICs resulting from the system's quantum interference processes. For this reason, we use the Landauer formalism and calculate the electronic transmittance, $\mathcal{T}(\omega)$. On the other hand, those two quantities only agree within the semiclassical Boltzmann limit. The inclusion of the electronic correlation that justifies the presence of the Kondo effect in the single electron transistor (SET) was derived in reference [61]. The QD was described by the Anderson impurity model, connected to the leads at different temperatures, within the Keldysh nonequilibrium

Green's function formalism. The transport coefficients $L_n$, Eq. B.9, were calculated, considering the particle current and thermal flux formulas through an interacting QD, [14–19]. In the present paper, the correlation in the system is "incorporated" into the electronic transmittance and computed by the CGFM, described in Appendix A.

Employing Eqs. B.4, B.7 and B.8, the dimensionless thermoelectric figure of merit, defined from Eq. 2, can be written as

$$ZT = \frac{\varepsilon}{1 - \varepsilon}. \tag{4}$$

It is clear from the above equation that the best dimensionless thermoelectric figure of merit $ZT$ occurs at the limit $\varepsilon \to 1$. In a previous paper [56], we showed that in the Kondo regime, the following equations are valid:

$$L_0(T^*) = -\left(L_0^S(T^*) - \frac{1}{h}\right)\cos(2\delta) + \frac{1}{h}, \tag{5}$$

$$\frac{L_1(T^*)}{T^*} = \frac{L_{01}^S(T^*)}{T^*}\sin(\delta)\cos(\delta), \tag{6}$$

and

$$\frac{hL_2(T^*)}{(k_B T^*)^2} = -\left(\frac{hL_2^S(T^*)}{(k_B T^*)^2} - \frac{\pi^2}{6}\right)\cos(2\delta) + \frac{\pi^2}{6}, \tag{7}$$

with $T^* = \frac{T}{T_K}$, being the temperature "normalized" by the Kondo temperature $T_K$, $L_0^S(T^*)$ is the coefficient $L_0(T^*)$ computed in the symmetric point of the model; an analogous situation exists for $\frac{hL_2^S(T^*)}{(k_B T^*)^2}$. These two functions, together with $\frac{L_{01}^S(T^*)}{T^*}$ are universal functions of $T^*$ and were computed employing NRG in a previous paper [56]. The parameter $\delta$ is the quantum phase shift associated with the Kondo scattering and "carries" the dependence with all the other parameters of the system.

Employing Eqs. 5, 6 and 7 in Eq. 3 we obtain

$$\varepsilon(T^*) = \frac{L_1^2(T^*)}{L_o(T^*)L_2(T^*)} = \sin^2(2\delta)\left[A(T^*)\cos^2(2\delta)B(T^*)\cos(2\delta) + C(T^*)\right]^{-1}, \tag{8}$$

where $A(T^*)$, $B(T^*)$ and $C(T^*)$ are universal expressions of $T^*$ given by:

$$A(T^*) = \frac{4\left[L_0^S(T^*) - \frac{1}{h}\right]}{L_{01}^2(T^*)} \times \left[\frac{hL_2^S(T^*)}{(k_B T^*)^2} - \frac{\pi^2}{6}\right](k_B T^*)^2, \tag{9}$$

$$B(T^*) = \frac{-4(k_B T^*)^2}{L_{01}^2(T^*)} \times \left[\left(L_o^S(T^*) - \frac{2}{h}\right)\frac{\pi^2}{6} + \frac{\frac{1}{h}L_2^S(T^*)}{(k_B T^*)^2}\right], \tag{10}$$

and

$$C(T^*) = \left(\frac{2\pi^2}{3h}\right)\frac{(k_B T^*)^2}{L_{01}^2(T^*)}. \tag{11}$$

Using $\sin^2(2\delta) = 1 - \cos^2(2\delta)$ in Eq. 8, and considering $\varepsilon(T^*) = 1$ we obtain a quadratic equation for $\cos(2\delta)$

$$\cos^2(2\delta)(A(T^*) + 1) + B(T^*)\cos(2\delta) + C(T^*) - 1 = 0. \tag{12}$$

To obtain the best thermal efficiency of the system on a temperature $(T^*)$, we must achieve $\varepsilon(T^*) = 1$, which means solving the equation in terms of the scattering phase shift,

$$\cos(2\delta) = \frac{-B(T^*) \pm D(T^*)}{2[A(T^*) + 1]}, \tag{13}$$

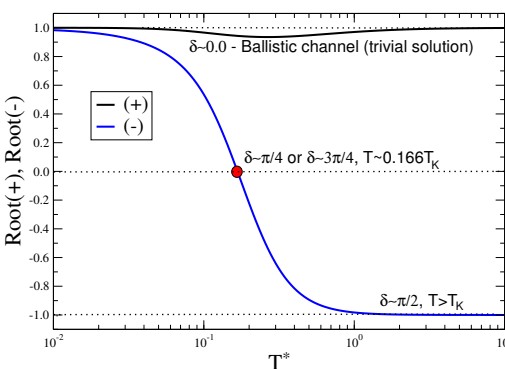

Figure 2: (Color online) Two possible "branches" of the right side of Eq. 13 vs. $(T^* = T/T_K)$. The dotted lines show the solutions of Eq. 13.

where $D(T^*)$ is a universal function of $(T^*)$ given by

$$D(T^*) = \sqrt{B^2(T^*) - 4[A(T^*) + 1][C(T^*) - 1]}. \tag{14}$$

Next, we calculate the limits of low and high temperatures of Eqs. 5, 6 and 7, analytically. Using these results, we show that in the first case exist two solutions for Eq. 13 associated with $\delta = \pi/4$ or $\delta = 3\pi/4$, at intermediate temperatures when the C=1, and in the limit of high temperatures exist only one solution with $\delta = \pi/2$. The universal functions in the limit of low temperatures $T^* \to 0$ have the following values: $L_0^S = 1$, $\frac{L_{01}^S}{T^*} = 0$, and $\frac{hL_2^S(T^*)}{(k_B T^*)^2} = \pi^2/3$. At high temperatures, in the limit when $T \to \infty$: $L_0^S = 0$, $\frac{L_{01}^S}{T^*} = 0$, and $\frac{L_2^S(T^*)}{(k_B T^*)^2} = 0$ [56].

Eq. 13 is valid in the Kondo regime, where the expected quantum phase scattering is $\delta \simeq \pi/2$ at low temperatures $T^* < 1$. However, in this regime, we found the two solutions ($\delta = \pi/4$ and $\delta = 3\pi/4$), as indicated in Fig. 2, that do not satisfy the Friedel sum rule, [59], and imply that is impossible to achieve the best thermoelectric efficiency for the SET.

In Fig. 2, we plot the two "branches", corresponding to the (+) and (−) solutions, for the right side of Eq. 13. The dotted horizontal lines are associated with possible solutions for the $\delta$ parameter. The dotted line in 1.0 is associated with a value of $\cos(2\delta) \simeq 1$, which presents a unique solution at $\delta \simeq 0.0$, that is a trivial solution and corresponds to having a ballistic channel in all the temperature ranges. When the "branch" associated with the root (−) is crossed by the dotted line at zero, at the red dot, two possible solutions are defined: $\delta \simeq \frac{\pi}{4}$ and $\delta \simeq \frac{3\pi}{4}$ at $T \sim 0.166 T_K$. For the bottom dotted line in −1.0, it is possible to obtain a unique solution $\cos(2\delta) \simeq -1$, $\delta \simeq \frac{\pi}{2}$ for high temperatures $T^* > 1$.

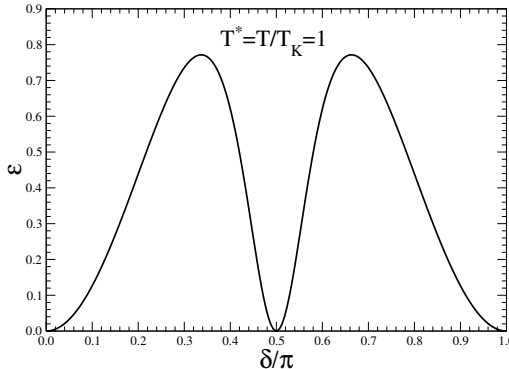

Figure 3: (Color online) The $\varepsilon$ parameter as a function of the quantum phase scattering $\delta/\pi$, for $T^* \simeq 1$.

Fig. 3 shows an estimation of the Mahan-Sofo parameter $\varepsilon$ as a function of $\delta/\pi$, for $T^* \simeq 1$. The $\varepsilon$ was computed employing the universal values for Onsager's coefficients, Eqs. 9-11, as a function of $T^*$ obtained in reference [56] and the Eq. 8. At around the values $\delta \simeq 0.33\pi \simeq \pi/4$ and $\delta \simeq 0.66\pi \simeq 3\pi/4$, values of $\varepsilon \simeq 0.8$ are obtained, which corresponds to a $ZT \simeq 4$. For $\delta = \pi/2$, $\varepsilon = 0$ corresponds to the model's symmetric point, where the Kondo resonance is present, and the thermopower vanishes [62].

The impossibility of attaining Carnot's efficiency for a SET in the Kondo regime in those systems does not invalidate the research of SETs as QD heat engines. One example of this is a recent work of a SET out of the equilibrium regime [63] that exhibits an efficiency in excess of 70% of the Carnot's efficiency while maintaining a finite power output. The experiment was performed in an out-of-equilibrium condition. However, the higher temperature difference between the hot ($T_h$) and the cold ($T_c$) reservoirs is given by $\Delta T = T_h - T_c \simeq 0.7K \simeq 7.0 \times 10^{-2}\Delta$, which is a small value that justifies the comparison with the results exposed in Fig. 3, obtained in the linear regime. Here, we employ an estimative of our energy unit $\Delta \sim 10K$ obtained by comparison of our theoretical results for electrical conductance at the unitary limit with experimental ones [62]. This experimental result [63] is in qualitative agreement with recent theoretical papers [64–66] that compute thermoelectric properties for a single QD system out of the equilibrium. They reported high thermoelectric performances when the QD has a single occupation, showing that the thermopower exhibits a change of sign due to the Kondo correlations at non-equilibrium conditions originated by asymmetrically tunneling to external electrodes. Those results are also compatible with the fact that the single SET does not exhibit Fano resonances near the chemical potential that could contribute to increases $ZT$ values. However, another promising system that could have its thermal efficiency increased is the metalloporphyrins QDs connected to gold leads, [47, 52, 54]. These systems exhibit Fano resonances near the chemical potential due to quantum interference processes from the symmetry-dependent coupling between molecular orbitals.

## 4 Two coupled identical quantum dots: Seeking for conditions that improve the thermoelectric efficiency

This section describes a system of two identical QDs (QD1 and QD2), a DQD system, as represented schematically in Fig. 4. Each QD exhibits a strong but finite local electronic correlation $U$. The QDs are connected by a conduction band (CB) through hybridizations $V_{QD1}$ and $V_{QD2}$ and immersed in a ballistic conduction channel, which is a physical situation adequate to originate the RKKY interaction. Two magnetic impurities, with localized spins, could achieve a ferromagnetic or antiferromagnetic order mediated by itinerant electrons. However, the RKKY interaction depends on the distance between the impurities and decreases as a $1/r^3$, where $r$ is the separation between the impurities (see Eq. 17.4 of the reference [67]). When we consider the two dots of the DQD system depicted in Fig. 4, they are well apart, so we described them using the GFs of two single QDs, and as a consequence, in this limit, the RKKY interaction between them vanishes. Therefore, the Hamiltonian of the DQD reduces to two Hamiltonians of the one QD as presented in Eq. 1 of the paper.

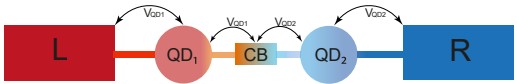

Figure 4: (Color online) Schematic picture of the DQD setup composed of two serial quantum dots, $QD_1$ and $QD_2$ immersed into conduction bands L, CB, and R, and subjected to the hybridizations $V_{QD1}$ and $V_{QD2}$.

In this section, we calculate an effective quantum phase shift $\delta_{eff}$ that could improve the thermoelectric efficiency and allow achieving the asymptotic Carnot's limit associated with the best thermoelectric efficiency. The $QD2$ is put in the electron-hole symmetry, while the gate voltage $E_d$ is varied in the $QD1$. The correlation in the localized states and the quantum scattering processes originated by the different regimes of the two dots causes an $\delta_{eff}$ analogous to the quantum scattering required to improve $ZT$, and it would be possible to enhance the thermoelectric efficiency.

Eq. 13 was obtained in the Kondo regime at low temperature and around the SIAM's symmetric point, which satisfies the particle-hole symmetry. The spin-flip scattering processes originate the Kondo peak in the density of states, which starts to "disappear" above the Kondo temperature due to thermal fluctuations. However, we can use Eq. 13 to approximate temperatures of the order of the $\Delta$ energy scale.

In the DQD setup, the transmission employed to compute the Onsager linear coefficients of Eq. B.9 is given by [68]

$$\mathcal{T}(\omega) = \Gamma^2 |G_{00}^{\sigma}(\omega)|^2, \tag{15}$$

where $\Gamma = \frac{V_2^2}{\Delta}$, $\Delta = \frac{\pi V_{QD2}^2}{2D} = 0.01$ is the Anderson parameter that defines the energy unit employed in all the calculations; and $D = 100\Delta = 1$ is the halfwidth of the conduction band. The local dressed Green's function is obtained from the setup of the DQD depicted in Fig. 4 and is given by

$$
\begin{aligned}
G_{00}^{\sigma}(\omega) &= \left[G_c^{\sigma}(\omega)\right]^3 V_{QD1}^2 G_{QD1}^{\sigma}(\omega) V_{QD2}^2 G_{QD2}^{\sigma}(\omega) \\
&= \left[G_{cond}^{\sigma}(\omega)\right]^2 V_{QD2}^2 G_{QD2}^{\sigma}(\omega) \\
&= i Im\left(G_{00}^{\sigma}(\omega)\right) + Re\left(G_{00}^{\sigma}(\omega)\right) \\
&= \left|G_{00}^{\sigma}(\omega)\right| e^{i\delta_{00}(\omega)},
\end{aligned}
\tag{16}
$$

where $G_{QD1}^{\sigma}(\omega)$ and $G_{QD2}^{\sigma}(\omega)$ are the correlated Green's functions associated with the first and second QD, respectively. These Green's functions are calculated through the CGFM, as presented in the appendix A. $G_c^{\sigma}(\omega)$ is the Green's function of the ballistic conduction channels, represented in the Fig. 4 by the labels "L", "CB" and "R", respectively. For simplicity, we use, in all the calculations, an uncorrelated rectangular conduction band of bandwidth $2D$

$$
\rho_c^{\sigma}(E_{\mathbf{k}\sigma}) = \begin{cases} \frac{1}{2D}, & \text{for } -D \leq E_{\mathbf{k}\sigma} - \mu \leq D, \\ 0, & \text{otherwise}, \end{cases}
\tag{17}
$$

with the corresponding GF

$$G_c^{\sigma}(\omega) = \frac{1}{2D} \ln\left(\frac{\omega + D + \mu}{\omega - D + \mu}\right). \tag{18}$$

The phase shift associated with the local GF is

$$\delta_{00}(\omega) = 2\delta_{cond}(\omega) + \delta_{QD2}(\omega). \tag{19}$$

$\delta_{cond}(\omega)$ is the phase shift associated with the "effective conduction channel" given by Eq. 21, and $\delta_{QD2}(\omega)$ is the QD2 phase shift set in the Kondo regime. These two path quantum interference processes can produce BICs and quasi-BICs (Figs. 6 and 12). The DQD system studied here behaves analogously to the geometrical configuration of a QD side coupled to a ballistic quantum wire (QW). Where it is clear the presence of a Fano resonance, characterized by the quantum interference of a "resonant" quantum scattering channel (the QD) and the "effective conduction channel" given by Eq. 21. The local phase shift associated with the quantum scattering process in the system can also be written as

$$\delta_{00}(\omega) = tan^{-1}\left(\frac{Im(G_{00}^{\sigma}(\omega))}{Re(G_{00}^{\sigma}(\omega))}\right). \tag{20}$$

The local "effective conduction channel" is described by the Green's function

$$G_{cond}^{\sigma}(\omega) = \left[(G_c^{\sigma}(\omega))^3 V_{QD1}^2 G_{QD1}^{\sigma}(\omega)\right]^{\frac{1}{2}}. \tag{21}$$

The DQD system can be seen as a single QD (QD2) "tuned" in the symmetric point and immersed in a conduction channel, with the leads renormalized given by Eq. 21. We can describe the DQD using the same Hamiltonian of the single QD of section 2, but using the local "effective conduction channel" given by Eq. 21. The average local quantum phase shift ("effective delta parameter", $\delta_{00}$) depends on the thermal fluctuations and the charge fluctuations on the first QD and can be written as

$$<\delta_{00}> = \int_{-D}^{D}\left(-\frac{\partial f(\omega, T)}{\partial \omega}\right)\delta_{00}(\omega)d\omega, \tag{22}$$

where, in all the calculations, Eq. 22 is used to calculate the averaged phase shifts. In particular, we refer to $\delta_{eff} = <\delta_{00}>$. The other two phase shifts will be referred as $\delta_{cond} = <\delta_{cond}>$ and $\delta_{QD2} = <\delta_{QD2}>$.

In the two next sections, we will explore, at low and intermediate temperatures, possible solutions of Eq. 13 for $\delta_{eff} \simeq \frac{\pi}{4}$ and $\delta_{eff} \simeq \frac{3\pi}{4}$. For temperatures above the $\Delta$ scale, with solutions at $\delta_{eff} \simeq \frac{\pi}{2}$, we will consider that Eq. 13 is approximately valid. We set the QD2 at the symmetric point of the SIAM as a scattered center and search for a quantum scattering interference process associated with charge fluctuations in the QD1, as pointed out in Eq. 19. In that case, it could originate an effective phase shift that improves thermoelectric efficiency, including the most interesting solution $cos(2\delta_{eff}) \simeq -1$, $\delta_{eff} \simeq \frac{\pi}{2}$ for temperatures above the temperature $\Delta$ scale. It is possible to obtain high $ZT$ values if $0 < \delta_{eff} \leq \frac{\pi}{2}$, with the solution at $\delta_{eff} \sim \frac{\pi}{4}$; or if $\frac{\pi}{2} \leq \delta_{eff} < \pi$, with the solution $\delta_{eff} \sim \frac{3\pi}{4}$.

## 5 Quasi-BICs linked to the enhancing process of $ZT$: Low temperature regime

The geometrical configuration of the QDs used in this paper is similar to the one employed in reference [69], where the authors considered two square cavities (two-dimensional quantum dots without any electronic correlation), serially connected by an infinite quasi-one-dimensional wire, where electrons have a continuous spectrum of energy. The appearance of BICs is controlled by the variation of the parameters associated with the QDs, their energy levels, and their spatial separation at zero temperature. Another interesting DQD system connected to a finite Majorana Kitaev chain was studied in reference [70], where the authors propose this system to encrypt Majorana fermions qubits as BIC states.

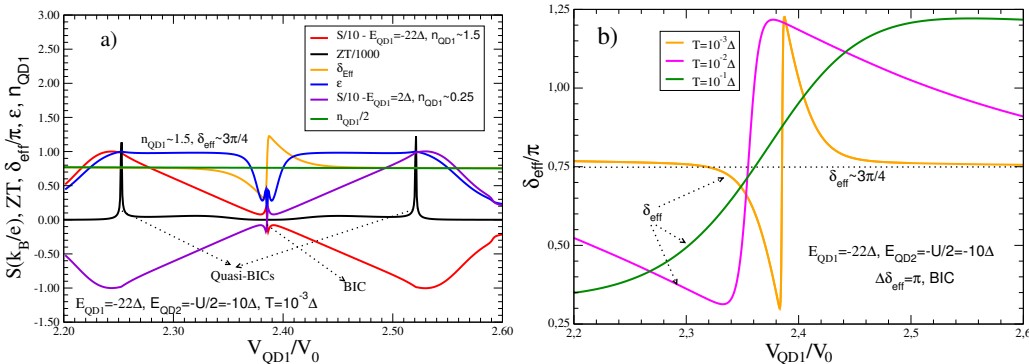

Figure 5: (Color online) Representative values of the QD1 hybridization to obtain quasi-BICs that manifest as strong peaks in the $ZT$. b) Detail of the $\delta_{eff}$ for different temperatures.

The system in the present paper is also composed of two QDs, serially connected by uncorrelated conduction bands, where electrons also have a continuous spectrum of energy given by Eq. 18. All the three conduction bands represented in the setup of Fig. 4 are ballistic leads and the scattering on the QDs are coherent resulting only in phase shifts of the electronic wave functions that are identified by the Friedel phase shift [71]. Buttiker et al. [72,73], showed that it is not correct to identify the Friedel phase with the phase of the amplitude of transmission because the transmission phase can depart from the Friedel phase and exhibit a nonanalytic behavior at points where the modulus of the transmission vanishes, while the Friedel phase remains continuous as a function of the energy. However, Orellana [74] showed that the Friedel phase in the presence of BICs is discontinuous as a function of the energy due to the delta-shaped character of the density of states as represented in Fig. 6(a,e).

To check the appearance of BICs in the DQD setup, we introduce an asymmetry in the hybridization at low temperatures, maintain fixed the hybridization $V_{QD2}$, and vary $V_{QD1}$. The interference process associated with the BICs is generally a single-particle interference effect. However, this effect can also appear in setups where the QDs exhibit electronic correlation effects $U_{12}$, as in the DQD geometry studied in reference [75], where the formation of BICs results from a quantum interference process that can be driven by the cross-correlation effects between the two QDs, $U_{12}$, and the asymmetries in dot-lead couplings. In the DQD studied in this paper, the electronic correlation, as well as the hybridization between the QDs, can act as driven parameters that generate Fano resonances at low temperatures, as plotted in Fig. 6, and multiple Fano resonances at higher temperatures, as exhibited in Fig. 12.

The unit of energy employed in the calculations is the Anderson parameter $\Delta = \frac{\pi V_{QD2}^2}{2D} = 0.01$, with $D = 100.0\Delta$, as defined below Eq. 15, which also furnishes the reference hybridization $V_0 = \sqrt{\frac{2D}{\pi}} = 0.798D$. We estimated the order of magnitude of $\Delta$, using experimental results of the electrical conductance $G$ at low temperatures. At the unitary limit in the Kondo regime for $GaAs(Al)$ [76] and $InAs(InP)$ [77] nanosystems, obtaining $\Delta \simeq 10K \simeq 10^{-3}$eV. At low temperatures, we fix the calculations at the temperature value $T = 0.01\Delta \simeq 0.1K$ because this temperature is sufficiently low for the system to show the presence of the Kondo effect. The remaining parameters for the QDs are the electronic repulsion $U_1 = U_2 = 20\Delta$, and the chemical potential $\mu = 0$. We fixed the localized energy levels of the QD2 in the particle-hole symmetric condition, $E_{QD2} = -\frac{U_2}{2} = -10\Delta$, whereas the localized level of the first QD, $E_{QD1}$ is allowed to vary.

In Fig. 5(a,b), we plot the representative values of the QD1 hybridization used in Fig. 6 to obtain quasi-BICs. In Fig. 5a) we calculate the thermopower $S$ for the two conditions with $E_{QD1} = -22.0\Delta$ and $E_{QD1} = 2.0\Delta$, that exhibit an anti-symmetrical shape as indicated

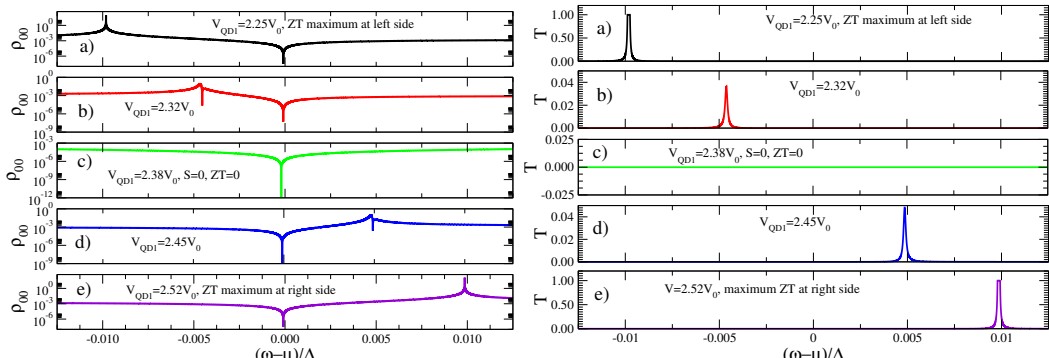

Figure 6: (Color online) Local density of states $\rho_{00}(\omega)$ and the transmittance $\mathcal{T}(\omega)$ vs. $\omega$ for representative values of $V_{QD1}/V_0$ as indicated in Fig. 5(a).

by the red and violet curves. The maximum $ZT$ values are attained for two hybridizations: $V_{QD1} = 2.25V_0$ and $V_{QD1} = 2.52V_0$; $ZT$ present huge peaks resulting from a quasi-BIC formation near $\mu = 0$, at $(\omega - \mu) \simeq \pm 10^{-2}\Delta$ (see Fig. 6(a,e)), which originates high values of the $\varepsilon$ parameter. The symmetrical condition, where $S = 0$, is attained when $V_{QD1}/V_0 = 2.38$ (the mean point between the two $ZT$ maxima). In Fig. 5b) $\delta_{eff}$ exhibits a "jump-discontinuity variation" of $\Delta(\delta_{eff}) \simeq \pi$ at the lowest temperature ($T = 10^{-3}\Delta$), evidencing the presence of a BIC in this point [73,74,78]. However, $\delta_{eff}$ is "smoothed" as the temperature increases. The same parameter set of Fig. 5(a, b), but with $E_{QD1} = 2.0\Delta$ produces similar results, but with $\delta_{eff} \simeq \pi/4$. We would like to emphasize that for $E_{QD1} = -22.0\Delta$, the occupation number of the dot QD1, is in the double occupation region with $n_{QD1} \simeq 1.5$; and when $E_{QD1} = 2.0\Delta$, the QD1 dot is in the empty orbital region $n_{QD1} \simeq 0.25$.

In Figs. 6 (a, b, c, d, e), we plot the local density of states $\rho_{00}(\omega)$ and the transmission coefficient $\mathcal{T}(\omega)$ vs. $\omega$ for different values of $V_{QD1}/V_0$, with $T = 0.01\Delta$ and $E_{QD1} = -22.0\Delta$. This figure shows the emergence of a single BIC, for $V_{QD1} = 2.38V_0$ (Figs. 6(c)), and two quasi-BICs for $V_{QD1} = 2.25V_0$ and $V_{QD1} = 2.52V_0$ (Figs. 6(a,e)), originating from the interaction of charge fluctuations in QD1 and the Kondo effect in QD2. For the $V_{QD1}/V_0 = 2.38$, the density of states $\rho_{00}(\omega)$ presents a sharp minimum close to $\omega = 0$ and the transmittance $\mathcal{T}(\omega) = 0$. On the other hand, varying the hybridization $V_{QD1}/V_0$ from this point to the maximum $\mathcal{T}(\omega)$ represented in graphs a) and e) the quasi-BICs emerge and reach their maximum intensities, producing enormous values of $ZTs$, as indicated in Fig. 5a).

Fig. 7 present thermoelectric properties computed for $V_{QD1} = 2.52V_0$, at low temperatures, when $E_{QD1} = -22.0\Delta$; $\varepsilon$ is close to the unit, $\varepsilon \simeq 1.0$, and achieves an efficiency at finite output

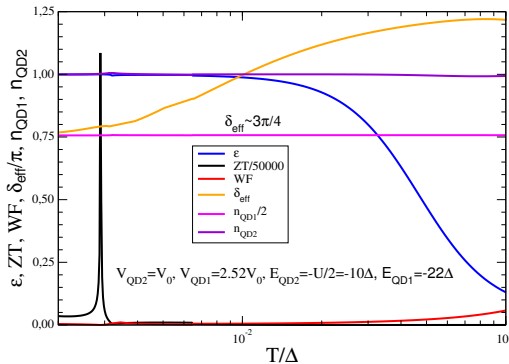

Figure 7: (Color online) The $\varepsilon$ parameter and related quantities, at low temperatures, for $V_{QD1} = 2.52V_0$.

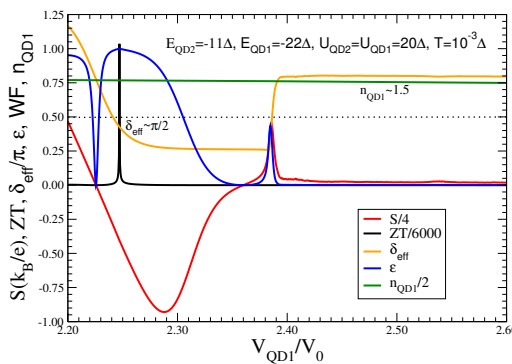

Figure 8: (Color online) Thermoelectric properties as a function of the QD1 hybridization, in a situation where the QD2 deviates slightly from the electron-hole symmetrical condition of Fig. 5a.

power, Eq. B.11, $\eta_{fo} \simeq \eta_C$, close to the Carnot efficiency $\eta_C$, for a large range of temperatures below $T \leq 4 \times 10^{-2}\Delta$. The $\delta_{eff} \simeq 3\pi/4$ and the Wiedemann-Franz law (WF) is not valid in regions with a huge ZT value, (no Wiedemann-Franz behavior (NWF))($WF = \frac{\kappa}{GL_N T} = \frac{3e^2\kappa}{TG\pi^2 k_B^2}$, $L_N$ is the Lorenz's number and $k_B$ is the Boltzmann's constant). However, this does not guarantee that a non-Fermi liquid behavior is present in those regions [79]. In the future, we intend to investigate the Fermi liquid behavior of the system and the possibility of the rising of NFL behavior in those regions of high ZT, where the WF law is not valid. This figure reveals that the Kondo effect present in the QD2 ($n_{QD2} \simeq 1$) is an important element that originates the conformation of the quantum-interference scattering process that enhances $ZT$ in a low-temperature range of approximately one order of magnitude. A similar result was obtained for the same parameters when $E_{QD1} = 2.0\Delta$; in this case, $\delta_{eff} \simeq \pi/4$, and the thermopower has a negative sign, indicating a hole thermal conduction processes.

In Fig. 8, we explore the behavior of the thermoelectric properties in a situation where the QD2 deviates slightly from the electron-hole symmetrical condition of Fig. 5a. We employed the following parameters: $E_{QD2} = -11.0\Delta$, and $U_{QD2} = U_{QD1} = 20.0\Delta$. The results show that the symmetry in Fig. 5(a) is loosed. In particular, we do not have the presence of a BIC associated with an "abrupt-jump" of $\delta \simeq \pi$ [73, 74, 78] (compare the $\delta_{eff}$ of Fig. 5(b) and Fig. 8). Only one huge ZT peak survives at $V_{QD1} \simeq 2.25V_0$, associated with the enhancement of the thermoelectric efficiency and linked to an effective quantum phase shift $\delta_{eff} \simeq \pi/2$. At $V_{QD1} \simeq 2.40V_0$ appears an "abrupt-jump" with $\delta_{eff} \simeq \pi/4$, with the Mahan-Sofo parameter presenting a peak with height close to $\epsilon \simeq 0.5$, but that does not have the $\delta \simeq \pi$ value, necessary to gives rise to a BIC conformation.

# 6 Quasi—BICs linked to the enhancing process of $ZT$: High temperature regime

We did not consider the lattice phonon effects in the calculations. However, from the experimental point of view, the phononic presence could be "reduced" by employing two 1D tunneling barriers to isolate the QDs from the electrodes. The material of these devices can be appropriately chosen to produce low $K_{ph}$ values [80]. The same effect could be obtained by the presence of an amorphous 1D conduction channel, where grain borders originate an "incoherent" contribution to the lattice thermal conductance, which act as scattering centers for phonons that contribute most strongly, reducing the thermal conductivity more than the

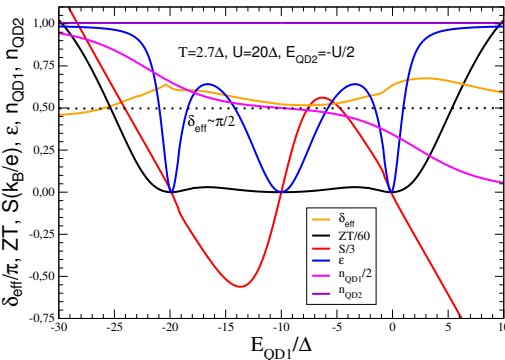

Figure 9: (Color online) Establishing the conditions to obtain an enhancement of the thermoelectric efficiency.

electrical conductivity [60, 81]. However, phonons could sometimes be desirable in the DQD geometry, as in the phonon-assisted transport study in a recent paper [36]. The authors construct a DQD detuned system employing tunnel barriers and controlling the tunnel couplings to separate the electronic and thermal transport, allowing the conversion of local heat into electrical power in a nanosized heat engine.

Fig. 9 presents the $\delta_{eff}$, $ZT$, $S$, $\varepsilon$, and the occupation numbers $n_{QD1}$ and $n_{QD2}$ vs. $E_{QD1}$ energy. As discussed in section 2, Eq. 13 exhibits a high-temperature solution when $\delta_{eff} \simeq \pi/2$, which is indicated by the dotted line in the figure. Furthermore, a $ZT \simeq 60$ for $E_{QD1} \simeq -30.0\Delta$ originates an $\varepsilon \simeq 1$, which implies, a high thermoelectric efficiency. The presence of the QD2 in the electron-hole symmetric condition, associated with the occupation number $n_{QD2} = 1$, and the charge fluctuations in the QD1, originates a quantum scattering process in the DQD system, analogous to the required to satisfy $\cos(2\delta) \simeq -1$ (see Fig. 2), at high temperatures, which enhances the thermoelectric efficiency.

The results corresponding to $E_{QD1} = -30\Delta$ are presented in Fig. 10, with the transport ruled by holes ($S > 0$). Similarly, results with the transport ruled by electrons ($S < 0$) is presented in Fig. 11 for $E_{QD1} = 10\Delta$. In Figs. 10 and 11, the "activation" of the quantum scattering process has three common elements: the existence of electron-hole symmetry in the QD2 scattering "center", charge fluctuations in the QD1, at temperatures $T \geq \Delta$, and an effective quantum phase shift $\delta_{eff} \simeq \pi/2$.

Fig. 10 shows some properties necessary to understand the enhancement of the thermoelectric efficiency as a function of temperature: $\delta_{eff}$, $\varepsilon$, $ZT$, $n_{QD1}$, $n_{QD2}$, and $S$. The charge

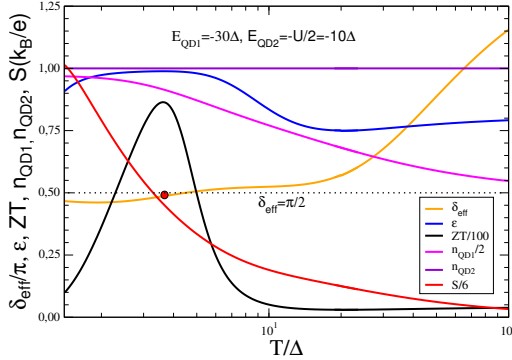

Figure 10: (Color online) The best conditions to enhance the thermoelectric efficiency considering, according to Fig. 9, $E_{QD1} = -30.0\Delta$. $S > 0$ indicates the transport by holes.

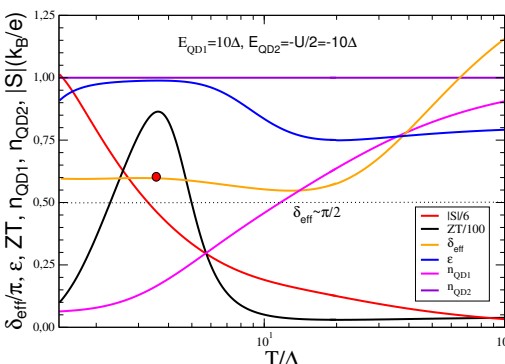

Figure 11: (Color online) The same of Fig. 10, but for $E_{QD1} = 10.0\Delta$. In this case, $S < 0$ indicates the transport by electrons.

fluctuations appear around the QD1, with its occupation number $n_{QD1}$, presenting a strong variation with the increase in temperature. At the same time, $QD2$ is maintained in the particle-hole symmetric situation, $n_{QD2} = 1$. This QD configuration results in the quantum scattering process that enhances the thermoelectric efficiency and is linked to charge fluctuations in the $QD1$. The highest value of $ZT \simeq 86.0$ is achieved at temperature $T = 3.66\Delta$, at around the red dot, when $\varepsilon \simeq 1$, and $\delta_{eff} \simeq \pi/2$. This setup produces an effective quantum scattering process that enhances the efficiency at finite output power, Eq. B.11, $\eta_{fo} \simeq \eta_C$, close to the Carnot efficiency $\eta_C$, for a large range of temperatures in the interval $\Delta > T > 10\Delta$. On the other hand, we do not have Kondo scattering processes in the system due to the high temperatures.

The same situation occurs in Fig. 11, for $E_{QD_1} = 10.0\Delta$, which is symmetrical to Fig. 10. The highest value of $ZT \simeq 86.0$ also occurs at temperature $T = 3.66\Delta$, with $\varepsilon \simeq 1$, and around the red dot; the $\delta_{eff}$ is close to $\pi/2$.

According to our results, the conditions that improve ZT are associated with a QD1 occupation close to the double occupation or the empty orbital regime. At low temperatures, as indicated in Fig. 5(a): When $E_{QD1} = -22.0\Delta$, $\delta_{eff} \simeq 3\pi/4$ and the dot QD1, is in the double occupation region with $n_{QD1} \simeq 1.5$, and when $E_{QD1} = 2.0\Delta$, the QD1 dot is in the empty orbital region $n_{QD1} \simeq 0.25$. At high temperatures, with $E_{QD1} = -30.0\Delta$, the ZT maximum in Fig. 10, is present when $\delta_{eff} \simeq \pi/2$ and $n_{QD1} \simeq 2$. Furthermore, when $E_{QD1} = 10.0\Delta$ as indicated in Fig. 11, ZT achieves its maximum when $\delta_{eff} \simeq \pi/2$ and $n_{QD1} \simeq 0.125$. In summary, in the conditions where ZT is enhanced, we do not expect an integer magnetic moment in the QD1, as the occupation of this dot is close to the double or empty orbital regime, which agrees with our initial supposition that the RKKY interaction must not be relevant in the DQD system studied in this paper.

In Figs. 12 (a, b, c) we plot the local density of states $\rho_{00}(\omega)$ and the transmittance $\mathcal{T}(\omega)$ vs. $\omega$ for different temperatures: a) $T = 1.4\Delta$, b) $T = 3.66\Delta$, and c) $T = 53.0\Delta$ and $E_{QD1} = -30\Delta$, $E_{QD2} = -10.0\Delta$ and $U = 20.0\Delta$. For $T = 1.4\Delta$ and $T = 3.66\Delta$, the effective quantum phase scattering is around $\delta_{eff} \simeq \pi/2$, and for $T = 53.0\Delta$, $\delta_{eff} \simeq \pi$.

The $\rho_{00}(\omega)$ exhibits two Fano-shaped structures located at $\omega - \mu = E_{QD1} = -30\Delta$ and $\omega - \mu \simeq E_{QD1} + U = -10\Delta$, that evolves with the increasing of temperature, producing a well-defined BIC at $\omega - \mu \simeq -22\Delta$ and two lateral quasi-BICs, as indicated in the $\rho_{00}(\omega)$ plot of Fig. 12c). However, as we expected, the presence of the BIC in $\rho_{00}(\omega)$ does not generate any signature in $\mathcal{T}(\omega)$. The increase of the temperature activates charge fluctuation in the QD1 from the many-body energy level $E_{QD1} + U = -10\Delta$ to the conduction channel, reducing the occupation number $n_{QD1}$, as indicated in Fig. 10 and, shaping the resonances at around the energies $E_{QD1} = -30\Delta$ and $E_{QD1} = -10\Delta$, as indicated in the $\mathcal{T}(\omega)$ plot of Fig. 12. Another effect observed in the plots is the spectral transfer of the states from the quasi-BIC at

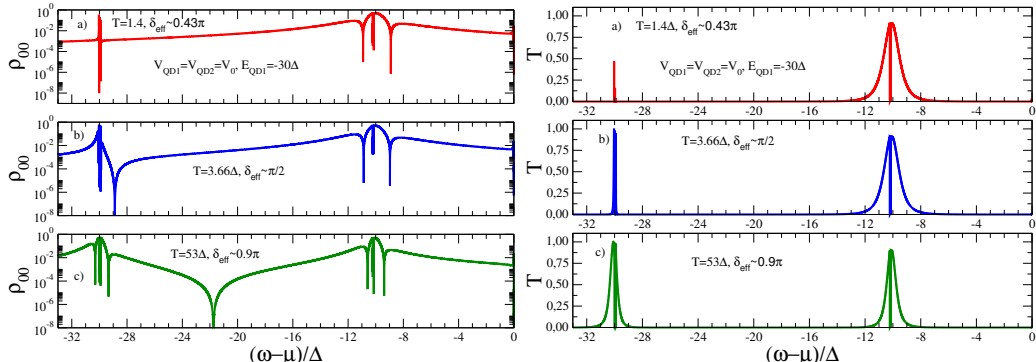

Figure 12: (Color online) Local density of states $\rho_{00}(\omega)$ and the transmittance $\mathcal{T}(\omega)$ vs. $\omega$, for different temperatures.

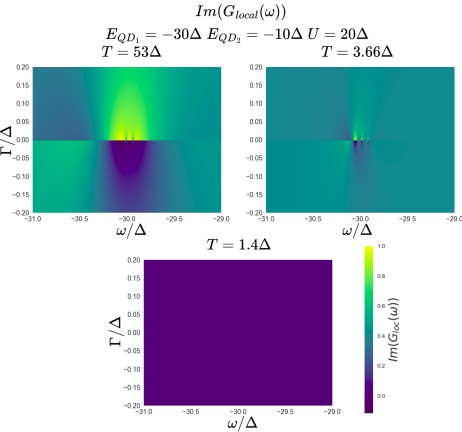

Figure 13: (Color online) Map of the density of states of the local Green function, showing the quasi-BIC formation at $E_{QD1} = -30.0\Delta$, for different temperatures.

$\omega - \mu = \simeq -10\Delta$ to the quasi-BIC formed at $\omega - \mu \simeq -30\Delta$. We identify the rising of these structures as linked with quasi-BICs generated by thermal excitations.

Fig. 13 shows the map of the density of states of the local GF $G_{00}(\omega + i\eta)$ (Eq. 16) in the complex plane, for $E_{QD1} = -30.0\Delta$ and temperatures $T = 53.0\Delta$, $T = 3.66\Delta$ and $T = 1.4\Delta$. The result presents a peak around $\omega = -30.0\Delta$ that increases its intensity as the temperature grows and is associated with the conformation of a quasi-BIC at this energy. The result is consistent with the one indicated in the $\mathcal{T}(\omega)$ plot of Fig. 12(a,b,c), where the width of the quasi-BIC peak increases with temperature.

Analogous results to those presented in Figs 12 and 13 were obtained when $E_{QD1} = 10\Delta$ at $(\omega - \mu) \simeq 10\Delta$ (not presented here). In this condition, the thermopower value changes its sign, and the results for the local density of states and the transmittance are a "mirror" with relation to the chemical potential value $(\omega - \mu) = 0.0$.

# 7 Conclusions and perspectives

The study of the universal thermoelectric properties for a QD immersed in a ballistic conduction channel in the Kondo regime, described by the SIAM, permits us to obtain the "ideal" quantum phase shift associated with a quantum scattering that increases the thermoelectric efficiency of

the system. We explored the possibility of enhancing the thermal efficiency of nanostructured materials, employing universal relations as a function of $T^* = \frac{T}{T_K}$. We calculated the ideal phase shifts to improve $ZT$ at the symmetric point of the SIAM, calculating the scattering phase shift using Eq. 13, that could enhance the thermoelectric efficiency of the device, allowing the system to achieve the best dimensionless thermoelectric figure of merit $ZT$. However, we showed that it is not possible to achieve this condition with a single QD device in the Kondo regime immersed in a ballistic quantum wire.

To overcome this limitation, we studied a DQD system with one of the QDs "tuned" into the electron-hole symmetric condition. We showed that the DQD exhibits an "effective" quantum phase shift $\delta_{eff}$ that improves the thermoelectric efficiency of the device, allowing to achieve the best dimensionless thermoelectric figure of merit $ZT$, at low (Sec. 5) and high temperatures (Sec. 6).

Our results indicate that the quantum scattering-interference process, associated with enhancing the thermoelectric efficiency of the DQD system, is linked with the rise of quasi-BICs. This process happens in two situations: At low temperatures, near the chemical potential, where the competition of the charge fluctuations in the QD1 and the Kondo effect present in the QD2 originates a Fano resonance associated with a large $ZT$ that produces a perfect conduction channel as indicated in Fig. 6. The other situation occurs at $T > \Delta$ temperature regime and generates quasi-BICs thermally activated but associated only with charge fluctuations in the QD1. The QD2 continues in the particle-hole symmetric point without charge fluctuations, but the Kondo effect has disappeared due to the high-temperature regime. Those quasi-BICs are present even at high temperatures $T \simeq 50\Delta$, a point that could be important for possible system applications in thermoelectric devices. However, It is necessary to work with temperatures that do not change the symmetry electron-hole in the QD2, i.e., that do not originate charge fluctuations in the QD2. Then, a considerable $U$ value is desirable.

The general conditions to generate BICs and quasi-Bics in the DQD setup, at low and high temperatures, have three common elements: The existence of electron-hole symmetry in one of the QDs scattering "centers" (QD2 here), an intense process of charge fluctuations in the other QD (QD1 here), and an effective quantum phase shift $\delta_{eff} \simeq 3\pi/4$ or $\pi/4$, for low temperatures and $\delta_{eff} \simeq \pi/2$ for high temperatures as derived in Fig. 2. However, there are some differences in the low and high-temperature quasi-Bics; the first is driven by the hybridization and the second by the temperature. Only one quasi-Bic exists at low temperatures, and we have twin quasi-Bic at high temperatures.

We expect that in the DQD system with localized energy values of the order of $E_{QD2}=-10.0\Delta \simeq -10^{-2}eV$, and interaction-repulsion energy $U = 2 \times |E_{QD2}| = 20.0\Delta \simeq 2 \times 10^{-2}eV$, the temperature values associated with the enhancing $ZT$ process would be $T > \Delta \sim 10K$. The numerical results show that the maximum temperature ($T_{max}$), where $ZT$ attains its maximum value is achieved approximately at $T_{max} \simeq \frac{3U}{2k_B} \times 10^{-1}$; here, $k_B$ is the Boltzmann's constant. However, if the $U$ parameter increases, the temperature associated with the enhancing quantum interference process of $ZT$ also increases. In order to obtain temperatures that enhance $ZT$ over 300K, we require $T_{max} \geq \frac{3U}{2} \times 10^{-1}\Delta \geq 300K$, $U \geq 2 \times 10^2 K \simeq 2 \times 10^{-2}$eV. We should employ systems that present strong "localization" tuned in the electron-hole symmetric point to originate a quantum scattering process that improves $ZT$ at higher temperatures, where the Carnot's thermoelectric efficiency is high.

We are working now to extend the results to the geometric configuration with the second QD in the symmetric condition but side-coupled to the conduction channel. The first step is extending the universal relations for the Onsager coefficients obtained in a previous paper [56], to the new geometric configuration and exploring the existence of analogous results. We expect the results of this paper to motivate experimental research to submit the experimental test of our theoretical predictions.

# Acknowledgments

**Funding information**    We are thankful for the financial support of the Research Division of the Colombia National University -Bogotá (DIEB). E. Ramos acknowledges support by Department of Science, Technology and Innovation by means of Colombian doctoral fellowship number 617-2. M. S. F. Acknowledges financial support from the Brazilian National Council for Scientific and Technological Development (CNPq) Grant. Nr. 311980/2021-0 and to Foundation for Support of Research in the State of Rio de Janeiro (FAPERJ) process Nr. 210 355/2018.

# A   Cumulant Green's function method

In this appendix we present a summary of the main equations necessary to establish the Cumulant Green's Functions Method (CGFM). The exact GF for the localized electrons with spin $\sigma$ can be exactly calculated though the $4 \times 4$ matrices equation (details can be found in reference [59], a brief presentation could be obtained in reference [62])

$$\mathbf{G}_\sigma(i\omega) = \mathbf{M}_\sigma(i\omega) \cdot (\mathbf{I} - \mathbf{A}_\sigma(i\omega))^{-1} \,, \tag{A.1}$$

where $i\omega$ are the Matsubara's frequencies, and from this equation follows

$$\mathbf{M}_\sigma(i\omega) = (\mathbf{I} + \mathbf{G}_\sigma(i\omega) \cdot \mathbf{W}_\sigma(i\omega))^{-1} \cdot \mathbf{G}_\sigma(i\omega), \tag{A.2}$$

where $\mathbf{M}_\sigma(i\omega)$ is the exact cumulant and $\mathbf{A}_\sigma(i\omega) = \mathbf{W}(i\omega) \cdot \mathbf{M}(i\omega)$. If the Hamiltonian is spin independent or commutes with the z component of the spin, all matrices can be reduced to two $2 \times 2$ matrices, being that $\mathbf{W}(z)$ matrix is defined by

$$\mathbf{W}_\uparrow(z) = |V|^2 \, \varphi_\uparrow(z) \begin{pmatrix} 1 & 1 \\ 1 & 1 \end{pmatrix}, \tag{A.3}$$

$$\mathbf{W}_\downarrow(z) = |V|^2 \, \varphi_\downarrow(z) \begin{pmatrix} 1 & -1 \\ -1 & 1 \end{pmatrix}, \tag{A.4}$$

where $z = \omega + i\eta$ corresponds to the analytic continuation of the Matsubara frequencies to the real axis, with $\eta = 0.0001$ is a small real quantity employed in the numerical calculations. Considering a rectangular band with half-width D we can write

$$\varphi_\sigma(z) = \frac{1}{2D} \, \ln\left(\frac{z - D + \mu}{z + D + \mu}\right). \tag{A.5}$$

The calculation of the exact cumulants correspond to the exact solution of the Hamiltonian, what is out of question here. Therefore, we introduce an approximation that consists in substituting the exact solution for an atomic cumulant that is defined by the atomic GF, that are calculated employing the Lehmann spectral representation

$$\begin{aligned}
\mathcal{G}^{at}_{\alpha\alpha'}(i\omega_s) = -e^{\beta\Omega} \sum_{n,r,r'} & \frac{\exp(-\beta\varepsilon_{n-1,r}) + \exp(-\beta\varepsilon_{n,r'})}{i\omega_s + \varepsilon_{n-1,r} - \varepsilon_{n,r'}} \\
& \times \langle n-1, r| \, X_{j,\alpha} \, |n, r'\rangle \langle n, r'| \, X^\dagger_{j,\alpha'} \, |n-1, r\rangle,
\end{aligned} \tag{A.6}$$

where $\Omega$ is the Thermodynamical potential and $\epsilon = E_d - \mu$ corresponds to the eigenvalue minus the chemical potential. This equation can be rewritten as

$$G^{at}_\sigma(z) = e^{\beta\Omega} \sum_{i=1}^{M} \frac{m_{i\sigma}}{z - u_{i\sigma}} \,, \tag{A.7}$$

where the $u_{i\sigma}$ and $m_{i\sigma}$ are the poles and the residues of the atomic GF of the localized electrons respectively.

The atomic solution of the exact Green's functions $\mathbf{G}_\sigma(z)$ follows the same rule as the equation A.1. Therefore, the atomic cumulants follows the equation

$$\mathbf{M}_\sigma^{at}(z) = \left(\mathbf{I} + \mathbf{G}_\sigma^{\mathbf{at}}(\mathbf{z}) \cdot \mathbf{W}_\sigma^o(\mathbf{z})\right)^{-1} \cdot \mathbf{G}_\sigma^{at}(z), \tag{A.8}$$

where the $\mathbf{W}_\sigma^o(z)$ is equal to

$$\mathbf{W}_\uparrow^o(z) = |\Delta|^2 \, \varphi_\uparrow^o(z) \begin{pmatrix} 1 & 1 \\ 1 & 1 \end{pmatrix}, \tag{A.9}$$

$$\mathbf{W}_\downarrow^o(z) = |\Delta|^2 \, \varphi_\downarrow^o(z) \begin{pmatrix} 1 & -1 \\ -1 & 1 \end{pmatrix}, \tag{A.10}$$

where $\Delta = \pi V^2/2D$ is the Anderson parameter that replace the hybridization in the atomic case. This normalization is needed because the use of the true hybridization $V$ superestimate the contribution of the conduction electrons, once we concentrate them in a single energy level $\epsilon_0$ where

$$\varphi_\sigma^o(z) = \frac{-1}{z - \varepsilon_o - \mu}, \tag{A.11}$$

is the GF of the zero-width band. The CGFM consists in substituting the atomic cumulant $\mathbf{M}_\sigma^{at}(z)$ in the exact equation for the full Green function, Eq. A.1. Details and the final expressions for all the GFs can be found in the reference [59].

# B Thermoelectric transport properties

In this paper we employ the thermoelectric transport properties for a QD immersed in a 1D ballistic conduction channel, -obtained in the linear regime-, essentially, we used the same equations (for the Onsager coefficients and thermoelectric properties) employed in reference [62]. A "summary" of its obtaining is presented here: To calculate the thermoelectric transport through the quantum dot in the steady-state condition, we applied a small external bias voltage $\Delta V = V_L - V_R$ and a small temperature difference $\Delta T = T_L - T_R$ between the left (L) and right (R) leads. In linear response theory, the current $J_{e,Q}$ can flow through the system under the action of temperature gradients $\vec{\nabla}T$ or electric fields $\vec{E} = -\vec{\nabla}V$. As in the system studied here, the chemical potential ($\mu = 0$) is fixed, and the contributions of the chemical potential gradients vanish ($\vec{\nabla}\mu = 0$). The charge current $J_e$ and the heat current $J_Q$ that cross the system are given by [82–84]:

$$J_e = e^2 L_0(T)(-\Delta V) + \frac{e}{T} L_1(T)(-\Delta T), \tag{B.1}$$

$$J_Q = e L_1(T)(-\Delta V) + \frac{1}{T} L_2(T)(-\Delta T), \tag{B.2}$$

where $e$ denotes the magnitude of the electrical charge and $L_0(T)$, $L_1(T)$, and $L_2(T)$ are the transport coefficients.

The electron conductance $G$ is measured under isothermal conditions $\Delta T = 0$. From Eq. B.1 we get

$$J_e = e^2 L_0(T)(-\Delta V), \tag{B.3}$$

and from the definition of electrical conductance [84, 85]

$$G(T) = -\lim_{\Delta V \to 0} \left. \frac{J_e}{\Delta V} \right|_{\Delta T=0} = e^2 L_0(T). \tag{B.4}$$

The electronic contribution to the thermal conductance $\kappa$ is usually measured by putting the sample on an open electrical circuit in such a way that $J_e = 0$ [83]. From Eq. B.1

$$(-\Delta V) = \frac{1}{eT}\frac{L_1(T)}{L_0(T)}(\Delta T). \tag{B.5}$$

Substituting Eq. B.5 into Eq. B.2 we get

$$J_Q = \frac{1}{T}\left(L_2(T) - \frac{L_1^2(T)}{L_0(T)}\right)(-\Delta T), \tag{B.6}$$

and from the definition of the thermal conductance [84, 85]

$$\kappa(T) = -\lim_{\Delta T \to 0}\frac{J_Q}{\Delta T}\bigg|_{J_e=0} = \frac{1}{T}\left(L_2(T) - \frac{L_1^2(T)}{L_0(T)}\right). \tag{B.7}$$

The thermopower (Seebeck effect) is defined by the relation [84, 85]

$$S(T) = \lim_{\Delta T \to 0}\frac{\Delta V}{\Delta T}\bigg|_{J_e=0} = \left(\frac{-1}{eT}\right)\frac{L_1(T)}{L_0(T)}. \tag{B.8}$$

The Boltzmann equation is only valid in the semiclassical regime, where the electrons behave as well-defined wave packets. Generally, this condition is satisfied at sufficiently high temperatures where the electron coherence is destroyed due to the strong inelastic scattering (diffusive transport). In this limit, the mean free path ($l$) is much larger than the electronic wavelength $\lambda$, ($l >> \lambda$). However, at low temperatures in the full quantum regime ($l << \lambda$), the electrons behave as waves (ballistic transport), the electron coherence is strong, and new phenomena arise that only a full quantum mechanics formalism can take into account (Landauer formalism for example), and the Boltzmann equation is not valid anymore. A comparison between the applicability limits of both methods can be found in the reference [86]. As examples of new full quantum phenomena [87], we have the Anderson localization, electronic transport in topological materials, far-from-equilibrium phenomena such as shot noise, and bound states in the continuum (BICs) as studied in this paper.

We compute the linear transport coefficients $L_0(T)$, $L_1(T)$, and $L_2(T)$, following the Dong and X. L. Lei paper [61]. They obtain the particle current and thermal flux formulas through an interacting QD connected to leads within the Keldysh non-equilibrium Green's functions (GF) framework. The transport coefficients are consistent with the general thermoelectric formulas derived earlier and are given by

$$L_n(T) = \frac{2}{h}\int\left(-\frac{\partial f(\epsilon, T)}{\partial \epsilon}\right)\epsilon^n \mathcal{T}(\epsilon, T)d\epsilon, \tag{B.9}$$

where $\mathcal{T}(\omega, T)$ is the transmittance for the electrons with energy $\epsilon = \hbar\omega$ and temperature $T$, here $h$ is the reduced Planck's constant and $f(\omega, T)$ is the Fermi-Dirac distribution function [84, 85].

On general grounds, we followed Mahan and Sofo's paper associating the efficiency to the $ZT$ parameter. However, we include a more detailed analysis of the efficiency at maximal output power (mo), $\eta_{mo}$, and at finite output power (fo) $\eta_{fo}$.

It is well known that Carnot's efficiency $\eta_C$ bounds the efficiency of all kinds of heat engines and is associated with a reversible engine with a zero production of entropy $\Delta S = 0$ and an infinity time for the working cycle to be completed. However, in "real" irreversible systems $\Delta S > 0$, the working cycle time is always finite. The study of $\eta_{mo}$ efficiency in

"real" irreversible systems has attracted much attention [88–90]. Curzon and Ahlborn [89] derived an upper bound limit, $\eta_{mo}$ efficiency, from an endoreversible Carnot heat engine: $\eta_{mo} = 1 - \sqrt{1 - \eta_C}$. On the other hand, the lower bound limit of the $\eta_{mo}$ efficiency that is relevant for the present work was obtained for fermionic thermochemical engines [90], with an infinitesimal width for the transmission coefficient $\Gamma \to 0$, and is given by [91]

$$\eta_{mo} \geq \eta_C/(1 + a(2 - \eta_C)), \tag{B.10}$$

with $a = 0.93593$. The $\eta_{fo}$ is given by

$$\eta_{fo} = \eta_C(\sqrt{ZT + 1} - 1)/(\sqrt{ZT + 1} + 1). \tag{B.11}$$

Recently, it was obtained experimentally a $\eta_{mo} \simeq 0.7\eta_C$, consistent with symmetric dissipation, in a QD system of InAs/InP [63].

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
