# Peer review of "Universality and the thermoelectric transport properties of a double quantum dot system: Seeking for conditions that improve the thermoelectric efficiency"

_SciPost Physics Core, doi:SciPost Phys. Core 7, 058 (2024)_

## Round 2 · Referee Report · Rosa Lopez (Referee 2) · 2024-7-7

Strengths

The paper offers a systematic investigation on the linear thermoelectrical transport properties on a serial double quantum dot it is analyzed when it can be reached the maximal thermoelectrical efficiency.

Weaknesses

The investigation is restricted only to the linear regime

Report

The authors have addressed correctly all the questions formulated by the referees and the manuscript has been improved greatly specially in the readability. Now the manuscript is more complete being a clear reference for serial double quantum dots and its thermoelectrical properties in the linear regime.

Recommendation

Publish (easily meets expectations and criteria for this Journal; among top 50%)

---

## Round 2 · List of Changes

Universality and the thermoelectric transport properties of a double quantum dot system: Seeking for conditions that improve the thermoelectric efficiency R. S. Cortes-Santamaria, J. A. Landazabal-Rodr´ıguez, J. Silva-Valencia, E. Ramos, M. S. Figueira and R. Franco. scipost − 202312 − 00043v1 To SciPost Team Thank the Scipost team for the email with the referee report about our paper. We also thank the referee for the physical analysis of the manuscript. We answered all the referee questions, improved the paper, and clarified the points raised in the report. We have thoroughly reviewed the paper, corrected the inconsistencies and repetitions, and fixed some typos. We include two new references, 67 and 79, in the new version of the paper. We send the source files, figures, and answers to the referees in a zip file. We also send an additional paper copy with all the corrections in red. We present a point-by-point response to the referees’ comments in the following. Answers to the referee “The work addresses the importance of designing new nanostructured systems presenting high thermoelectrical efficiencies. The treatment for calculation of such efficiencies is somehow novel as long as it relates the figure of merit to the scattering phase shifts. The authors identify the main mechanisms to have a large efficiency and show that the proposed system, a double quantum dot, is able to reach extraordinary ZT values even in the high temperature regime.” Answer: We thank the referee for reading the paper carefully and considering the work relevant to the nanoscopic thermoelectric research area. “The employed numerical technique to solve the Hamiltonian is discussed in the manuscript however it is not compared with other techniques.” The following text in red in the last paragraph of Section 2: “In this work, we employed the exact results obtained in an earlier paper of our group, using the NRG technique based on the universality associated with the Onsager coefficients [Ref. 56]. In Sec. 3, we derive the dimensionless thermoelectric figure of merit ZT using NRG. However, to obtain the results presented in the paper, we employed the cumulant Green’s function method (CGFM) for the SIAM summarized in Appendix A, [Ref. 59]. The CGFM describes the different regimes of SIAM well: Empty orbital, mixed-valence, and Kondo, and it led to results that agreed with the NRG predictions.” “The thermoelectrical analysis is done solely for the linear regime although a calculation for the efficiency at maximum power is presented in the Apendix.” Answer: The referee is right. The work’s focus is on the linear regime, which delimits its scope. “Only situations where one of the quantum dots is in the symmetry situation is presented and for me it is not clear why the authors discard a more general scenario.” Answer: To consider the referee’s suggestion, we included a new figure on page 13, now Figure 8, and renumbered the others. In Figure 8, we calculate the thermoelectric properties as a function of the QD1 hybridization in a situation where the QD2 deviates slightly from the electron-hole symmetrical condition of Fig. 5a. We include the following text in red to discuss the new figure: “In Fig. 8, we explore the behavior of the thermoelectric properties in a situation where the QD2 deviates slightly from the electron-hole symmetrical condition of Fig. 5a. We employed the following parameters: EQD2 = −11.0∆, and UQD2 = UQD1 = 20.0∆. The results show that the symmetry in Fig. 5(a) is loose. In particular, we do not have the presence of a BIC associated with an “abrupt jump” of the effective quantum phase shift of π (see references 73,74,78) (compare the δef f of Fig. 5(b) and Fig. 8). Only one huge ZT peak survives at VQD1 ≃ 2.25V0, associated with the enhancement of the thermoelectric efficiency and linked to an effective quantum phase shift δef f ≃ π/2. At VQD1 ≃ 2.40V0 appears an “abrupt jump” with δef f ≃ π/2, with the Mahan-Sofo parameter presenting a peak with height close to ϵ ≃ 0.5, but that does not have the π value, necessary to gives rise to a BIC conformation.” We also discuss the relationship between the occupation number of the QD1 dot and the phase shift. We slightly modify Figure 5a and its corresponding discussion to include the information on the occupations of the QD1 dot. We include in red the following text after the discussion of Fig. 5 (a,b) on page 13. We would like to emphasize that for EQD1 = −22.0∆, the occupation number of the dot QD1, is in the double occupation region with nQD1 ≃ 1.5; and when EQD1 = 2.0∆, the QD1 dot is in the empty orbital region nQD1 ≃ 0.25. We also include in red the following text after the discussion of Fig. 10 on page 13. According to our results, the conditions that improve ZT are associated with a QD1 occupation close to the double occupation or the empty orbital regime. At low temperatures, as indicated in Fig. 5(a): When EQD1 = −22.0∆, δef f ≃ 3π/4 and the dot QD1, is in the double occupation region with nQD1 ≃ 1.5, and when EQD1 = 2.0∆, the QD1 dot is in the empty orbital region nQD1 ≃ 0.25. At high temperatures, with EQD1 = −30.0∆, the ZT maximum in Fig. 10, is present when δef f ≃ π/2 and nQD1 ≃ 2. Furthermore, when EQD1 = 10.0∆ as indicated in Fig. 11, ZT achieves its maximum when δef f ≃ π/2 and nQD1 ≃ 0.125. In summary, in the conditions where ZT is enhanced, we do not expect an integer magnetic moment in the QD1, as the occupation of this dot is close to the double or empty orbital regime, which agrees with our initial supposition that the RKKY interaction must not be relevant in the DQD system studied in this paper. “The authors present an interesting work that nicely addresses the importance of having nanosystems working as highly efficient thermoelectrical devices. For such purposes, the authors indicate that quantum dots could perform such a job. Indeed, QDs are known to be good thermoelectrical systems due to their strong dependence on energy for transmission. With this aim the authors analyze two situations, the first corresponds to a single QD in which the thermoelectrical efficiency cannot achieve the maximal value. The way to show this result is through the scattering amplitude. Since this system is not a good candidate the authors introduce another QD connected by tunneling to the other QD via a continuous channel. It is natural to have interference phenomena. Under certain conditions the transmission displays peaks due to the formation of localized states and due to this phenomenon the thermoelectrical power enhances a lot. However, here I have some questions” “1. The Kondo effect is needed since it introduces an energy scale denoted by the Kondo temperature that makes these localized states to have a very narrow width and this is responsible for having the large ZT. If this is right, the authors should clarify better why Kondo resonances are needed in the interference phenomenon to lead to such high ZT values.” Answer: To answer the referee’s question, we include the following text in red on page 3 of the Introduction: “All calculations are based on the universality associated with the Onsager coefficients described in reference [56]. This universality is linked to a quantum scattering center modeled by the SIAM in the electron-hole symmetric condition, valid for the limit of low energy excitations associated with the Kondo effect. This universality permits us to obtain the effective quantum phase shifts that improve the thermoelectric efficiency. The other condition is a quantum interference process that permits achieving the effective quantum phase shifts required and described in section 3.” “2. It is not clear to me why the authors neglect the RKKY interaction that can appear in this system.” Answer: We thank the referee for calling our attention to this point. We modify the first paragraph of section 4, on page 8, introducing the following text in red: “Each QD exhibits a strong but finite local electronic correlation U. The QDs are connected by a conduction band (CB) through hybridizations VQD1 and VQD2 and immersed in a ballistic conduction channel, which is a physical situation adequate to originate the RKKY interaction. Two magnetic impurities, with localized spins, could achieve a ferromagnetic or antiferromagnetic order mediated by itinerant electrons. However, the RKKY interaction depends on the distance between the impurities and decreases as a 1/r3 , where r is the separation between the impurities (see Eq. 17.4 of the reference [67]. When we consider the two dots of the DQD system depicted in Fig. 4, they are well apart, so we described them using the GFs of two single QDs, and as a consequence, in this limit, the RKKY interaction between them vanishes. Therefore, the Hamiltonian of the DQD reduces to two Hamiltonians of the one QD as presented in Eq. 1 of the paper.” “3. The continuum states connecting the two dots could lead, for some system parameters the emergence of a non-Fermi liquid behavior. Can the authors comment on this?.” Answer:We consider this observation relevant, and during the numerical calculations, we discussed this point. We decided to refrain from advancing the study of Fermi-liquid/non-Fermi-liquid behavior in the system to delimit the scope of the work. We include the following text on page 12 of section 5: “The δef f ≃ 3π/4 and the Wiedemann-Franz law (WF) is not valid in regions with a huge ZT value, (no Wiedemann Franz behavior (NWF))(W F =κGLN T =3e2κT Gπ2k2B LN is the Lorenz’s number and kB is the Boltzmann’s constant). However, this does not guarantee that a non-Fermi liquid behavior is present in those regions, as indicated in reference 79. In the future, we intend to investigate the Fermi liquid behavior of the system and the possibility of the rising of NFL behavior in those regions of high ZT, where the WF law is not valid.” “It is needed to have one of the quantum dots in the symmetric case? What it can happen if the two dots are gated?.” Answer: Maintaining the second QD “tuned-pined” in the symmetric point is crucial. The universality linked to the symmetric condition loses applicability if it is outside the symmetric point. In all the paper calculations, we look for conditions that improve ZT by employing information about the quantum phase shift obtained, assuming the presence of a quantum scattering center in the symmetric condition. Please see the new Figure 8 on page 13. “Could the authors discuss the main differences in relation of the thermoelectrical properties this work has in comparison with other works dealing with double dots. For instance in DQDs in parallel Fano factor can also be present.” Answer: We agree with the referee that in DQDs, the Fano factor is present in the parallel configuration. We also include the following text in red on page 17 of the section on Conclusions and Perspectives: “We are working now to extend the results to the geometric configuration with the second QD in the symmetric condition but side-coupled to the conduction channel. The first step is extending the universal relations for the Onsager coefficients obtained in a previous paper (ref[56]) to the new geometric configuration and exploring the existence of analogous results.” Best regards, Roberto Franco and coauthors. Note: This is the text attached in the previously answer to the referee. All the new parts in the paper are in red color in the new version re-submitted now. The new version of our paper is at (https://arxiv.org/abs/2302.09099), arXiv:2302.09099v2 .

---

## Editorial Decision

published